# Predicting crystal form stability under real-world conditions

Dzmitry Firaha[1✉], Yifei Michelle Liu[1✉], Jacco van de Streek[1], Kiran Sasikumar[1], Hanno Dietrich[1], Julian Helfferich[1,13], Luc Aerts[2], Doris E. Braun[3], Anders Broo[4], Antonio G. DiPasquale[5], Alfred Y. Lee[6], Sarah Le Meur[2], Sten O. Nilsson Lill[4], Walter J. Lunsmann[7], Alessandra Mattei[8], Pierandrea Muglia[7], Okky Dwichandra Putra[9], Mohamed Raoui[10], Susan M. Reutzel-Edens[11,14], Sandrine Rome[2], Ahmad Y. Sheikh[8], Alexandre Tkatchenko[12], Grahame R. Woollam[10] & Marcus A. Neumann[1✉]

The physicochemical properties of molecular crystals, such as solubility, stability, compactability, melting behaviour and bioavailability, depend on their crystal form[1]. In silico crystal form selection has recently come much closer to realization because of the development of accurate and affordable free-energy calculations[2–4]. Here we redefine the state of the art, primarily by improving the accuracy of free-energy calculations, constructing a reliable experimental benchmark for solid–solid free-energy differences, quantifying statistical errors for the computed free energies and placing both hydrate crystal structures of different stoichiometries and anhydrate crystal structures on the same energy landscape, with defined error bars, as a function of temperature and relative humidity. The calculated free energies have standard errors of 1–2 kJ mol$^{-1}$ for industrially relevant compounds, and the method to place crystal structures with different hydrate stoichiometries on the same energy landscape can be extended to other multi-component systems, including solvates. These contributions reduce the gap between the needs of the experimentalist and the capabilities of modern computational tools, transforming crystal structure prediction into a more reliable and actionable procedure that can be used in combination with experimental evidence to direct crystal form selection and establish control[5].

Molecular crystals are important components of food products[6], semiconductors[7], explosives[8], agrochemicals[9] and pharmaceuticals[10–12]. Their physicochemical properties depend on an interplay of chemical composition and molecular packing within a crystal structure, known as a crystal form or polymorph when more than one arrangement exists[10,11,13,14]. The development of an undesirable crystal form can have harmful consequences, as shown in the cases of ritonavir and rotigotine[12,15,16]. Crystal form selection remains a challenge because of the implications for physical and chemical stability, solubility, dissolution, nucleation barriers, mechanical properties, filtration, powder flow and, under consideration here, the formation of hydrates and solvates stable within accessible temperature and humidity ranges[1,17–19]. To complement experimental efforts, computational methods, in particular crystal structure prediction (CSP), are becoming important for polymorph risk assessment and control[3,5,20–23]. The capabilities of CSP have improved greatly in recent years by the inclusion of temperature-dependent free-energy calculations, yet without rigorously assessing the accuracy of such predictions[2,4,24–31]. CSP can be used to identify the most stable, possibly still to be discovered, crystal

form of pharmaceutically relevant molecules[20,21,32] and has started to be applied early in the molecular and materials design cycles by balancing accuracy and computational efficiency[33]. Furthermore, CSP is used in the prediction of stoichiometric hydrates[34] and solvates[35,36], but without explicitly considering relative humidity or solvent activity. Free-energy calculations have been used to construct hydrate–anhydrate phase diagrams, still requiring experimental calibration for every compound and pair of crystal forms[37]. A data-driven and topological algorithm[38] has been recently used in conjunction with CSP for the prediction of fractional or nonstoichiometric hydrates to evaluate the ever-present risk of hydrate formation for industrially relevant compounds, due in part to the ubiquity of water vapour in the atmosphere. The current study addresses four of the most salient open issues: the need for more accurate and affordable free-energy calculations, the lack of a reliable free-energy benchmark, the quantification of computational errors and the prediction of stability relationships between hydrates and anhydrates as a function of temperature and relative humidity without a need for compound-dependent experimental calibration. These advances bridge the gap between the capabilities of in silico crystal

[1]Avant-garde Materials Simulation, Merzhausen, Germany. [2]UCB Pharma SA, Chemin du Foriest, Braine-l'Alleud, Belgium. [3]Institute of Pharmacy, University of Innsbruck, Innsbruck, Austria. [4]Data Science and Modelling, Pharmaceutical Sciences, R&D, AstraZeneca Gothenburg, Mölndal, Sweden. [5]Genentech, South San Francisco, CA, USA. [6]Merck, Analytical Research & Development, Rahway, NJ, USA. [7]GRIN Therapeutics, New York, NY, USA. [8]Solid State Chemistry, Research & Development, AbbVie, North Chicago, IL, USA. [9]Early Product Development and Manufacturing, Pharmaceutical Sciences R&D, AstraZeneca Gothenburg, Mölndal, Sweden. [10]Novartis Pharma, Basel, Switzerland. [11]Cambridge Crystallographic Data Centre, Cambridge, UK. [12]Department of Physics and Materials Science, University of Luxembourg, Luxembourg City, Luxembourg. [13]Present address: JobRad, Freiburg, Germany. [14]Present address: SuRE Pharma Consulting, Zionsville, IN, USA. ✉e-mail: dzmitry.firaha@avmatsim.eu; yifei.michelle@gmail.com; marcus.neumann@avmatsim.eu

form selection and the needs of bench practitioners, as demonstrated by case studies on two pharmaceutical compounds.

## Composite free-energy calculations

Ideally, free-energy calculations would be carried out using a single, accurate ab initio method and the standard machinery of statistical thermodynamics. In practice, however, such an approach is economically unfeasible. Time and cost are vital elements for innovators, whether they use purely experimental methods or complement with CSP. The requirement to treat many crystal forms in a single CSP study limits the acceptable amount of central processing unit (CPU) time required for a single free-energy calculation to about one day on 1,000 cores. An alternative is to combine a variety of affordable calculations that capture various physical effects and, when united, correct for the shortcomings of each other.

As shown in Extended Data Fig. 1, our free-energy calculation method TRHu(ST) 23 (an acronym for temperature- and relative-humidity-dependent free-energy calculations with standard deviations) combines the composite PBE0 + MBD + $F_{vib}$ approach[4,39] (where PBE0 is a hybrid functional composed of the Perdew–Burke-Ernzerhof (PBE) functional with 25% Hartree–Fock exchange energy, MBD is many-body dispersion energy, and $F_{vib}$ is the free energy of phonons at finite temperature) with an additional single-molecule correction[29] and reduces the CPU time requirements of the phonon calculations by blending force field and ab initio calculations. Moreover, imaginary and very soft vibrational modes, hydrogen-bond stretch vibrations and methyl-group rotations are explicitly sampled (Methods).

## Free-energy benchmark

For calculated energies to be used in process design and risk assessment, knowledge of the associated errors is as important as the predicted values themselves. The quantification of these errors has received almost no attention in CSP because an extensive and reliable benchmark for solid–solid free-energy differences of industrially relevant compounds was not available. The scientific literature describes solid–solid phase transformations and lists stability relationships, but usually the published data do not constitute the determination of a free-energy difference between two determined crystal structures. For example, the free-energy differences between polymorphs can be obtained from their solubility ratio in infinite dilution, but a dataset is only complete when the crystal structures of both polymorphs have been solved and the solubilities of both polymorphs have been measured in a common solvent, where both forms are only moderately soluble. Literature information is often incomplete, possibly because there was no incentive for the authors to publish all measurements, but more likely because the experiments are usually very challenging for metastable polymorphs. Assuming all data were complete for a compound, it may be split over several publications by different authors, complicating matters further in case the naming of polymorphs or the assignment of structural and thermodynamic properties is open to interpretation.

For this work, complete data for a chemically diverse and industrially relevant set of compounds have been collected from the literature and from several experimental contributors working in academia and industry. Apart from the 12 free-energy differences obtained from solubility ratios, four reversible (enantiotropic) phase transitions between polymorphs and 21 reversible hydrate–anhydrate phase transitions as a function of relative humidity have been used to determine free-energy differences. At the phase-transition temperature that separates the stability domains of two polymorphs, the free energies are equal by definition. Therefore, every experimental observation of a reversible phase transition constitutes the measurement of a free-energy difference that is zero. The case of hydrate–anhydrate phase transitions is discussed separately below. Our free-energy benchmark is described in detail in the Supplementary Information. The performance of our free-energy calculations on the benchmark is documented in Extended Data Tables 1 and 2.

## Transferable error estimation

To apply a quantitative risk assessment to compounds, it is necessary to express the deviation between the experimental and the computed free-energy differences in terms of a small set of parameters that enable extrapolation of the observed errors to chemical compounds not part of the benchmark, accounting for molecular size and chemical variability. We rationalize the observed energy discrepancies in terms of standard deviation ($\sigma$) of the energy error per water molecule, $\sigma_{H_2O} = 0.641$ kJ mol$^{-1}$ and standard deviation of the energy error per atom, $\sigma_{at} = 0.191$ kJ mol$^{-1}$, for non-water atoms in the compound (Supplementary Information). Standard deviations of free energies and their differences can be derived from these basic values for any chemical compound and water content using the formulae presented in the Methods based on Gaussian error propagation. For example, the standard error for a molecule consisting of $N$ atoms is $\sqrt{N}$ times larger than $\sigma_{at}$, and using this relationship the standard error per water molecule can be translated to a standard error per atom in a water molecule of $\sigma_{H_2O}/\sqrt{3} = 0.379$ kJ mol$^{-1}$. When compounds from the benchmark did not fulfil all quality criteria (Supplementary Information), they were excluded from the determination of $\sigma_{H_2O}$ and $\sigma_{at}$. In general, one would expect from a well-calibrated error estimation model that the deviation between the computational and experimental free-energy differences normalized by the expected standard error should follow a Gaussian distribution with a standard deviation of one. Extended Data Fig. 2 shows that this expectation is fulfilled for the benchmark compounds.

## Hydrate–anhydrate phase transitions

Hydrate–anhydrate phase transitions are different from the other experimental sources of free-energy information discussed above. Water molecules leave the solid state on dehydration and have to be modelled explicitly in their liquid or gas phase. Calculating the gas-phase free energy of water (Supplementary Information), we established that a systematic underestimation of the phase-transition relative humidity can be avoided by adding an empirical correction, $\mu_{H_2O,corr}^{o} = -1.77$ kJ mol$^{-1}$, to the computed gas-phase chemical potential of water. The correction was fitted with $\sigma_{H_2O}$ (Supplementary Information). Figure 1 shows the experimental and calculated phase-transition relative humidities (top, right scale), which are related to the pressure-dependent part of the chemical potential (bottom, left scale). The experimental relative humidities are reproduced to within a factor of 1.7 on average over all compounds in the validation set. Without the chemical potential correction for water, calculation and experiment still agree within a factor of 2.4 (Extended Data Fig. 3), proving that the chemical potential correction is not strictly required, thus making our approach directly applicable to other solvents.

## Pharmaceutical case studies

We present two pharmaceutical case studies—radiprodil and upadacitinib—to illustrate the predictive power of our method featuring the new standard graphical representation of CSP results, with defined error bars, as a function of temperature and relative humidity. The structures of the two molecules are shown in Fig. 2.

Radiprodil is an NR2B-negative allosteric modulator initially developed for the treatment of neuropathic pain[40]. More recently, it has shown potential in the treatment of infantile spasms[41] and may provide therapeutic benefits in paediatric epileptic disorders[42].

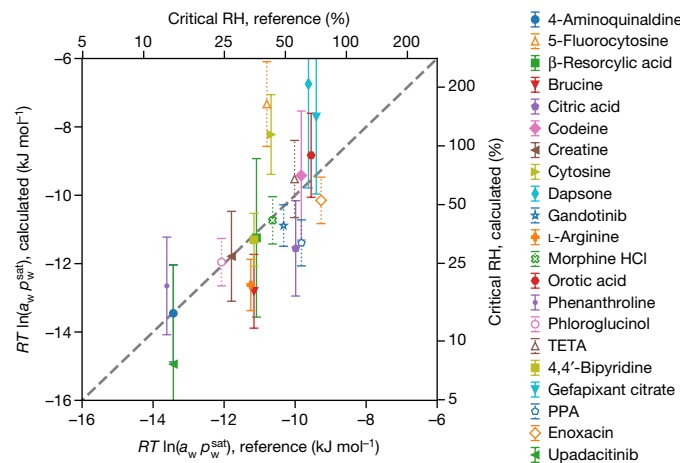

**Fig. 1 | Calculated versus reference pressure-dependent part of the chemical potential of water at phase transition.** The relative humidity (RH) is shown on the secondary axes. Reference systems excluded from the calculation of statistical errors because of ambiguities in experimental data are shown with open markers and dotted error bars. Error bars represent 1 standard error. $R$ is the universal gas constant, $T$ is the absolute temperature, $a_w$ is water activity and $p_w^{sat}$ is the saturated vapor pressure of water. PPA, pipedimic acid; TETA, triethylenetetramine dichloride.

The crystal-energy landscape of radiprodil, computed at defined temperatures and relative humidities, is shown in Fig. 3. The full symbols correspond to a predicted anhydrate, monohydrate or dihydrate crystal structure characterized in terms of its free energy and density.

The experimental anhydrate, monohydrate and dihydrate forms, indicated by open symbols in Fig. 3, correspond to the most stable predicted crystal structures for each of the respective stoichiometries, demonstrating the accuracy of our composite energy calculation method. Error bars are provided for each predicted crystal structure. The energy differences between the structures on the lower part of the crystal-energy landscape are of the same order of magnitude as the error bars. At the top of the crystal-energy landscape, the error bars are larger, because not all energies are calculated at the highest level of theory to preserve CPU time.

The temperature dependence of selected anhydrate structures, including the experimental forms A and C, is shown in Fig. 4.

The error bars are translated into stripes in Fig. 4, with the accuracy in the prediction of the phase-transition temperature being shown by the extent of the crossover region of the stripes. Here the predicted phase-transition temperature of 481.60 K differs from the experimental value of 343.15 K by 138.45 K, which is less than $1\sigma$ of 183 K. At every temperature, all free energies have been shifted such that they average to zero. Without this temperature-dependent shift, the free-energy curves would be hardly distinguishable (Extended Data Fig. 4).

Upadacitinib is a Janus kinase inhibitor that works by blocking certain signals causing inflammation. Upadacitinib is being developed for a range of auto-immune diseases[43], and to date it has been approved by the US Food and Drug Administration for rheumatoid arthritis,

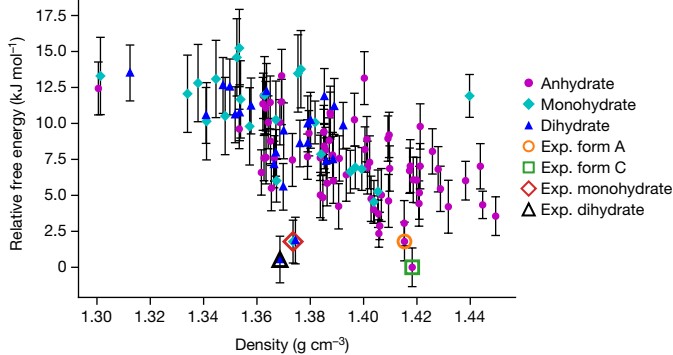

**Fig. 3 | Free-energy landscape of radiprodil hydrate and anhydrate forms at 298.15 K and a relative humidity of 50%.** Predicted structures of anhydrate, monohydrate and dihydrate of radiprodil and experimentally observed (exp.) structures of anhydrate form A, anhydrate form C, monohydrate and dihydrate.

psoriatic arthritis, ulcerative colitis, atopic dermatitis and ankylosing spondylitis[44].

The crystal-energy landscape of the anhydrate, hemihydrate and monohydrate structures of upadacitinib is shown in Extended Data Fig. 5. The two experimentally observed phases, form I (hemihydrate) and form III (anhydrate) again correspond to the most stable predicted structures at their respective stoichiometries. The free energies of the most stable predicted anhydrate, hemihydrate and monohydrate at 25 °C as a function of relative humidity are shown in Fig. 5.

No monohydrate has been observed experimentally, fitting with the fact that the monohydrate is predicted to be less stable than both the anhydrate and the hemihydrate at 25 °C over the full range of relative humidities. The predicted phase transition from form III to form I at 7.8% relative humidity is found experimentally at 14% relative humidity, well within the $1\sigma$ confidence interval ranging from 2.4% relative humidity to 26% relative humidity. Varying both temperature and relative humidity, the solid–solid phase diagram indicating the most stable form as a function of the thermodynamic variables can be constructed as shown in Fig. 6. The more complex solid–solid phase diagram of radiprodil is presented in Supplementary Fig. 15.

## Discussion

### Impact on crystal form selection

Both case studies are examples of well-controlled polymorphic systems, in which the respective stable crystal forms at specified temperatures

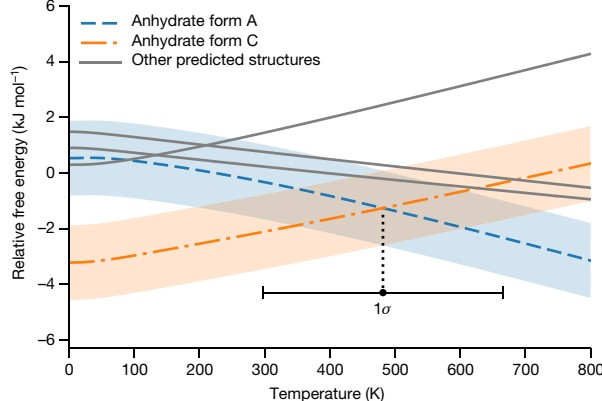

**Fig. 4 | Relative free energy versus temperature for the anhydrate forms of radiprodil, showing the transition at 481.6 K.** The $1\sigma$ confidence interval on the transition temperature is shown below the free-energy curves. The experimental transition is measured at 343.15 K.

**Fig. 2 | 2D diagrams of case-study molecules.** Radiprodil (left) and upadacitinib (right).

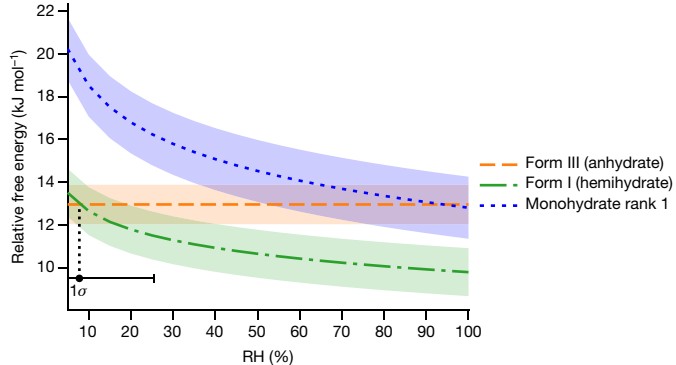

**Fig. 5 | Calculated free-energy versus relative-humidity curves of form III (anhydrate), form I (hemihydrate) and the predicted rank 1 monohydrate of upadacitinib at 25 °C.** The rank 1 monohydrate is predicted to be unstable for the entire humidity range. The $1\sigma$ confidence interval on the relative humidity at coexistence between form I and form III is shown beneath the free-energy curves. The experimental transition is observed at 14% relative humidity at 25 °C.

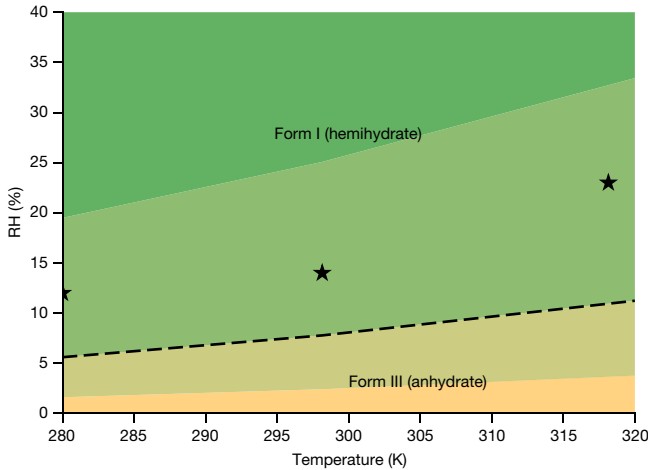

**Fig. 6 | Predicted phase diagram of upadacitinib.** The dashed line indicates the predicted phase boundary, with shading to indicate the $1\sigma$ confidence intervals. The experimentally measured phase-transition points are indicated by stars.

and relative humidities have been prepared experimentally. However, it has been suggested[45] that for about 30% of the compounds in late-phase pharmaceutical development, the most stable crystal form does not readily crystallize. Modern crystal-energy landscapes reveal these missing forms, and using relative energy differences and calculated errors, we can assess the risk that a more stable predicted form is actually more stable in the real world, with the predicted magnitude of decrease in solubility should the new form crystallize. If the missing form is perceived as a risk and further efforts are made to bring it into existence, the a priori predicted phase diagram shows the experimental conditions at which the missing form is thermodynamically favoured. If, on the contrary, it has not been possible to obtain the missing form despite intensified experimental screening, the same predicted phase diagram provides a map for the design of a robust process to mitigate the risk of encountering the missing form.

### Current versus required accuracy

If CSP is to be universally incorporated into industrial processes in, for example, pharmaceutical development, properties such as temperature and relative humidity of phase transitions must be predicted with an accuracy similar to or better than the experimental values. This is especially true for phenomena occurring within the range of conditions likely to be encountered in processing and storage. However, for many thermodynamic properties, even small free-energy errors give rise to relatively large confidence intervals in predictions because of, for example, the exponential dependence of those measurable quantities on free-energy differences or the rather similar slopes of the temperature-dependent free-energy curves of organic crystals. For most of the reference compounds, the standard error of the free-energy difference, $\sigma_{\Delta F}$, is between 1 kJ mol$^{-1}$ and 2 kJ mol$^{-1}$. At the current level of accuracy, hydrate–anhydrate phase-transition relative humidities are predictable to within a factor of 1.7 and the $1\sigma$ error of 183 K obtained for the anhydrate–anhydrate phase-transition temperature of radiprodil is representative of what has been observed in numerous confidential contract research studies. Therefore, at present, our method enables the prediction of probable phase transitions between two crystal forms, although not the point at which it will occur. The prediction of phase-transition temperatures to within 10 K requires a further improvement in accuracy by a challenging factor of 20.

### Rigorous quantification of uncertainty

With error bars representing a standard deviation under normal distribution, Gaussian statistics enable us to quantify the

reliability of the predictions. For example, for a pair of polymorphs with a temperature-dependent reversible phase transition (enantiotropic relationship), the overlap of the $1\sigma$ free-energy error can be translated into a confidence interval of the transition temperature.

Comparing the magnitudes of the errors $\sigma_{at}$ and $\sigma_{H_2O}$, we see that the error per atom of the water molecule at 0.379 kJ mol$^{-1}$ is approximately two times larger than the error per non-water atom at 0.191 kJ mol$^{-1}$. Both values are affected by the computational and the experimental errors, and as such represent the upper limits of the actual computational error. The larger error per atom for water is consistent with the fact that water is generally considered difficult to model. By contrast, 70% of the atoms of the compounds in our test set are carbon atoms or their covalently bonded neighbours that are much easier to describe. The difference of a factor of two between the two errors suggests a substantial dependence of the error per atom on the atomic species. Therefore, the derived value for $\sigma_{at}$ is an average that applies only to compounds that are close to the average chemical composition of the benchmark.

### Outlook

Borrowing some words from a famous quote of John Maddox[46], despite substantial recent progress, it is "one of the continuing scandals in the physical sciences that it remains in general impossible to predict" the solid–solid phase-transition temperatures of pharmaceutical compounds to within a few kelvins.

The availability of diverse and reliable data is a prerequisite for further improvements. To establish the benchmark presented in this work, a collaboration of a substantial number of companies and academic groups was required. It is important that academic groups are able to apply for funding related to the measurement of accurate structural and thermodynamic properties of organic solids, and this attempt should not be considered outdated in thermodynamics by the funding agencies. Likewise, it would be desirable that pharmaceutical companies all around the world declassify more of their internal solid-state data when compatible with Intellectual Property requirements. With more data, the present work could be extended to all relevant crystallization solvents, and atom-species-dependent atomic errors could be determined, thus enabling a finer description of the computational errors.

For a further reduction of the computational error, two broad approaches can be identified. On the one hand, two important physical phenomena have not been taken into account in the present work—namely, thermal lattice expansion and a full treatment of the

contribution of anharmonicity to the lattice free energy. Because ab initio methods are expensive and the force fields are not accurate enough for such an endeavour, it can be expected that machine learning force fields will provide a cost-accuracy compromise to capture these effects[47,48]. On the other hand, there is the accuracy of the ab initio calculations themselves. The single-point energy-correction scheme of our method can, in principle, provide consistent energies and forces for lattice-energy minimization and vibrational sampling. The single-point energy-correction scheme itself may then be taken to a higher level of theory, calculating lattice energies directly at the CCSD(T)[49] level or performing monomer, dimer[50] and, potentially, trimer corrections at that level of theory.

We hope that our work will inspire others to tackle the organic solid–solid phase-transition temperature challenge.

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

## Methods

### Computational details

**Naming.** The name of our energy calculation method is TRHu(ST) 23, an abbreviation of temperature- and relative-humidity-dependent free-energy calculations with standard deviations, and it should be pronounced as Trust 23. The name is meant to encompass both the actual energy calculations and the model for transferable error estimates calibrated on a specific benchmark. Because the energy calculations, the error estimation model and the benchmark will continue to evolve, we have added the year of publication to refer to our implementation and to define a naming scheme for improvements to come.

**Electronic structure energy corrections.** All experimental crystal structures were minimized with the Perdew–Burke–Ernzerhof (PBE) functional[51] augmented with Neumann–Perrin dispersion correction (PBE-NP)[52] with the light basis set with the 2010 species defaults using the FHI-aims ab initio package[53–57]. Subsequently, a set of single-point energy corrections and a monomer correction were applied to the minimized structures. Single-point energy corrections included a light-to-tight basis-set correction, a functional correction from PBE to PBE0[58,59] and a correction from PBE-NP to PBE with a non-local many-body dispersion correction (PBE-MBD-NL)[39,60,61]. The monomer correction followed the same protocol as described elsewhere[29] and was performed in addition to the aforementioned single-point corrections with second-order Møller–Plesset theory with a correction for van der Waals dispersion (MP2D)[62] using the NAO-VCC-4Z[63] basis set in FHI-aims and the anaconda psi4/mp2d module[64,65]. The performance of the method with various components removed, such as the single-point corrections or the single-molecule MP2D correction, is shown in Supplementary Table 24 and Supplementary Figs. 16–19 and demonstrates that the combined method improves performance substantially over previously published methods. All calculations, including those with third-party code, were carried out in GRACE[66].

**Phonon calculations.** Before the second-order dynamic matrix (Hessian) calculation, a cell replication to a supercell was carried out for all structures to guarantee at least a distance of 8 Å between each atom and its nearest symmetry copy. Next, a Cartesian displacement of 0.01 Å was applied in six Cartesian directions to compute the forces at the PBE-NP level and to derive the Hessian from finite differences.

For the determination of eigenvalues and eigenmodes, a $k$-point-dependent Hessian of the original cell was first derived from the supercell Hessian for every $k$-point compatible with the periodic boundary conditions of the supercell. Subsequently, the mass-weighted Hessian was used to obtain eigenvalues, from which the vibrational contribution to the free energies was computed using the harmonic approximation. Imaginary modes corresponding to double-well potentials were observed at the PBE-NP level for only one form of gaboxadol HCl and one form of verebecestat. The handling of imaginary modes is described below.

**Eigenmode following corrections.** A few more corrections to the harmonic approximation were added to the free energies. The corrections are termed the imaginary mode correction and the very soft mode correction and include explicit mode sampling from modes with eigenvalues less than 0 cm$^{-1}$ and 25 cm$^{-1}$, respectively. These modes were explicitly sampled with at least three points in each direction (positive and negative) and approximated to a fourth-order polynomial potential. Next, the free energy of each mode was calculated by explicitly solving the one-dimensional Schrödinger equation for the polynomial potential. The free-energy contribution was splined out

for modes between 15 cm$^{-1}$ and 25 cm$^{-1}$ with a fourth-order polynomial. The same procedure without splining was applied to imaginary modes, explicitly sampling the double-well potential and solving its Schrödinger equation.

**Methyl top correction.** For the methyl top correction, we applied a standard approach, that is, explicit methyl top rotation potential sampling inside the crystal, then solved the Schrödinger equation for a particle on the computed potential.

**Hydrogen-bond correction.** The hydrogen-bond correction aimed to correct the zero-point vibration energy for hydrogen atoms in X−H···Y, where X, Y = N, O, F. Each hydrogen-bonded hydrogen atom was displaced along six Cartesian directions by 0.01 Å to compute a local mass-weighted Hessian matrix. Solving an eigenmode problem for this matrix gives three eigenvalues. The eigenvector with the largest eigenmode of the hydrogen-bonded hydrogen atom was followed in the positive and negative directions to sample the potential, with at least two points in each direction. Energy levels used in the free-energy calculations were obtained from a solution of the Schrödinger equation for a particle on the quartic potential fitted for sampled points.

**Large-cell correction.** To conserve CPU time, phonon calculations at the ab initio level are limited to very small supercells that are not large enough to capture the effect of phonon band structure on the lattice free energy, in particular for acoustic modes. Therefore, a large-cell correction was carried out using tailor-made force fields[67] reparametrized with additional reference data at the PBE-NP level of theory for the force calculations. Using the phonon calculations described above, including imaginary mode and soft mode corrections, force field lattice free energies were computed for the small supercell already used in the ab initio phonon calculations and a larger supercell with minimal distances of 24 Å between symmetry copies of an atom. The difference between the two lattice free-energy calculations was used as a correction. As already described above, eigenvalues and eigenmodes were explicitly calculated for every $k$-point compatible with the periodic boundary conditions of the supercell. This way the band structure of all modes, including acoustic modes, is explicitly taken into account. The contribution of the three acoustic modes at the gamma point was approximated by $3RT$. It is important to note that imaginary modes can only be explicitly sampled and evaluated if an explicit supercell is available for the corresponding $k$-point.

For the gas-phase water calculations, the large-cell correction was not applied.

### Statistical model of error

The ultimate aim of crystal structure prediction as part of solid form selection is the derisking of the developed form to prevent the recurrence of a disaster such as that of ritonavir. As such, the uncertainty or expected error of the calculated values must be carefully quantified relative to experimental observables. The error in our calculations can be attributed to myriad sources, such as errors from the density functional, the dispersion correction, the choice of basis set, the harmonic approximation for the computation of the vibrational partition function and basis-set superposition errors. Because we wanted to evaluate if the interactions of the water molecule within the crystal give rise to larger errors than other atoms on average, we separated the total error into two contributions: one contribution per atom for the main compound in the asymmetric unit cell and one contribution per water molecule in the asymmetric unit. Each error is assumed to be normally distributed and statistically independent, such that the error for a single free energy per organic molecule would also be normally distributed, with a variance found by Gaussian error propagation:

$$\sigma^2(F - \hat{F}) = \frac{m}{N}\sigma_{at}^2 + \frac{n}{N}\sigma_{H_2O}^2 \qquad (1)$$

Here, $F$ and $\hat{F}$ are the observed and predicted free energies of the crystal structure per organic molecule, $m$ is the number of atoms per organic molecule, $n$ is the number of water molecules per organic molecule and $N$ is the number of organic molecules per asymmetric unit. $\sigma_{at}$ and $\sigma_{H_2O}$ are the standard deviations of the per-organic-molecule-atom and per-water-molecule errors, respectively. Further details are provided in the Supplementary Information.

Likewise, the total error on the calculated energy difference between two crystal structures is also normally distributed with

$$\sigma^2(\Delta F - \Delta\hat{F}) = \sum_{i=1,2}\left(\frac{m}{N_i}\sigma_{at}^2 + \frac{n_i}{N_i}\sigma_{H_2O}^2\right) \qquad (2)$$

$\Delta F$ and $\Delta\hat{F}$ are the observed and predicted free-energy differences between two crystal structures of the same organic molecule, indexed by $i$. The parameters $N$ and $n$ may now be different for each crystal structure.

$\sigma(F - \hat{F})$ and $\sigma(\Delta F - \Delta\hat{F})$ refer to $1\sigma$ of the error for the free energy of a single crystal structure and the free-energy difference between two crystal structures, respectively, and are referred to as $\sigma_F$ and $\sigma_{\Delta F}$. Both quantities are per organic molecule.

### Chemical potential correction of water

A single fitted correction on the chemical potential of water, $\mu_{H_2O,corr}^\circ$, is needed to account for the reference state of the free-energy calculations and other factors such as the neglect of basis set superpositioning errors and thermal lattice expansion. The performance of the method without water chemical potential correction is shown in Extended Data Fig. 3.

### Standard error calculation

A validation dataset of anhydrate solubility ratios and enantiotropic phase transitions (Supplementary Data) is used to independently solve for $\sigma_{at}$. A validation dataset of hydrate–anhydrate phase-transition systems (Supplementary Data) is used to solve for $\sigma_{H_2O}$, and $\mu_{H_2O,corr}^\circ$ simultaneously. Further details on the error model and characterizing the error distributions are provided in the Supplementary Information.

### Anhydrates free-energy differences

Starting from solubility ratios and reversibly determined anhydrate phase transitions, a reference free-energy difference $\Delta F_{ref}$ was calculated at the experimental temperature. This reference free-energy difference was compared with the predicted free-energy difference at the same temperature. $\sigma_{at}$ was calculated using Gaussian error propagation. Further details are provided in Supplementary Table 22 and Supplementary Fig. 12.

### Hydrate–anhydrate phase transitions

For the hydrate systems, the reference data point used was the relative humidity at hydrate–anhydrate coexistence, or critical water activity, measured at 298.15 K. To compute the standard error on the predicted coexistence relative humidity, the standard deviation of the error of the free-energy difference is normalized to one molecule of water ($\Delta n$ is the difference in the number of water molecules between the two structures):

$$\sigma_{wat} = \frac{\sigma(\Delta F - \Delta\hat{F})}{\Delta n} \qquad (3)$$

Note the difference between $\sigma_{wat}$ in equation (3) and $\sigma_{H_2O}$ from equations (1) and (2); the latter is a constant, whereas the former is system-dependent. Further equations for the calculation of the

predicted phase transition from calculated hydrate–anhydrate free energies are provided in the Supplementary Information.

### Crystal structure prediction

Crystal structure predictions were carried out with GRACE 2.7 following the procedure described previously[32].

## Data availability

Crystal structure data, crystal solubilities, phase-transition temperatures and relative humidity values of the phase transitions used in this paper are provided in the Supplementary Information and Supplementary Data.

## Code availability

The free-energy calculations described in this work were originally implemented in and carried out with v.3.0 of the commercial GRACE software package that can be licensed from Avant-garde Materials Simulation (AMS). As agreed with the editor, the actual free-energy calculations have been extracted from GRACE and collected as source code or pseudo-code into a library that is available from AMS upon request. The library requires the user to provide a SuperCellManager object that performs the actual single-point energy calculations. It has been tested that the library provides the same results as the corresponding code in GRACE.

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

**Acknowledgements** D.E.B. acknowledges funding from the Austrian Science Fund (FWF projects T593-N19 and V436-N34). M.A.N., D.F., K.S., J.H., H.D., J.v.d.S and Y.M.L. acknowledge funding from the German "Zentrales Innovationsprogramm Mittelstand" (projects EP180192 and ZF4716901DF9).

**Author contributions** M.A.N. and D.F. conceived the project and developed the method. D.F. developed the validation protocol. D.F., J.H. and M.A.N. wrote the software infrastructure to carry out the calculations, with contributions from K.S. and H.D. J.v.d.S. compiled the initial validation set and provided the crystallography, including disorder analysis and inspecting and correcting crystal structures. D.F., Y.M.L., J.v.d.S. and K.S. ran most of the calculations. Y.M.L. and D.F. analysed the validation results. A.G.D. contributed to the calculations. A.T. contributed to the method development. L.A., D.E.B., A.B., A.Y.L., S.L.M., S.O.N.L., W.J.L., A.M.,

P.M., O.D.P., M.R., S.R., A.Y.S. and G.R.W. contributed to the experimental data. S.M.R.-E. assisted with the writing of the paper. Y.M.L. wrote the first version of the paper with input from all authors. M.A.N. rewrote the text after the reviewers' comments.

**Competing interests** A.M. and A.Y.S. are employees of AbbVie and may own AbbVie stock. G.R.W. is an employee of Novartis and may own Novartis stock. M.A.N. is the owner of AMS. D.F., Y.M.L., J.v.d.S., K.S., H.D. and J.H. are or have been employees of AMS and have no other competing interests.

**Additional information**
**Correspondence and requests for materials** should be addressed to Dzmitry Firaha, Yifei Michelle Liu or Marcus A. Neumann.

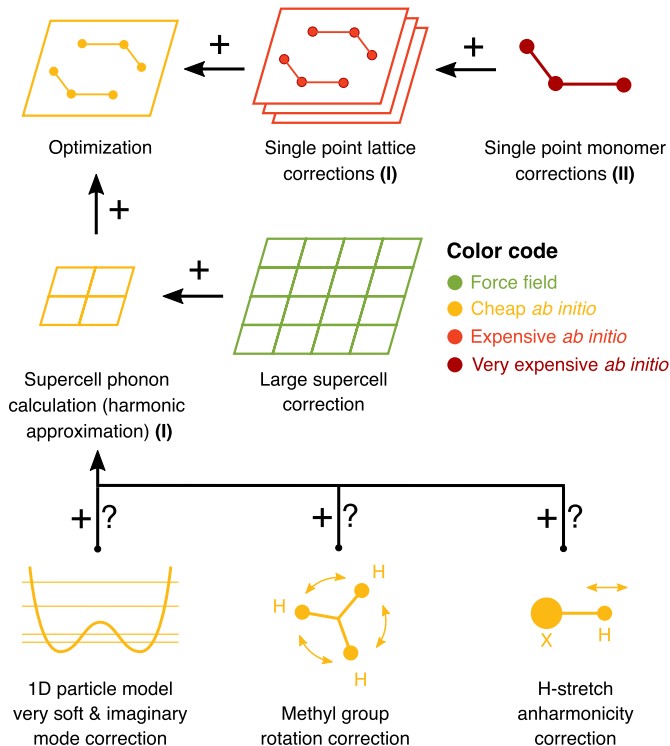

Optimization

Single point lattice
corrections **(I)**

Single point monomer
corrections **(II)**

**Color code**
- Force field
- Cheap *ab initio*
- Expensive *ab initio*
- Very expensive *ab initio*

Supercell phonon
calculation (harmonic
approximation) **(I)**

Large supercell
correction

1D particle model
very soft & imaginary
mode correction

Methyl group
rotation correction

H-stretch
anharmonicity
correction

**Extended Data Fig. 1 | Schematic of the free energy method introduced in this work.** Bold Roman numerals indicate methods following (I) Hermann et al.[39] and Hoja et al.[4], and (II) Greenwell, et al.[29]. Full details are presented in the Supplementary Information. For contributions with a question mark the current validation data set is too small to confirm a positive impact.

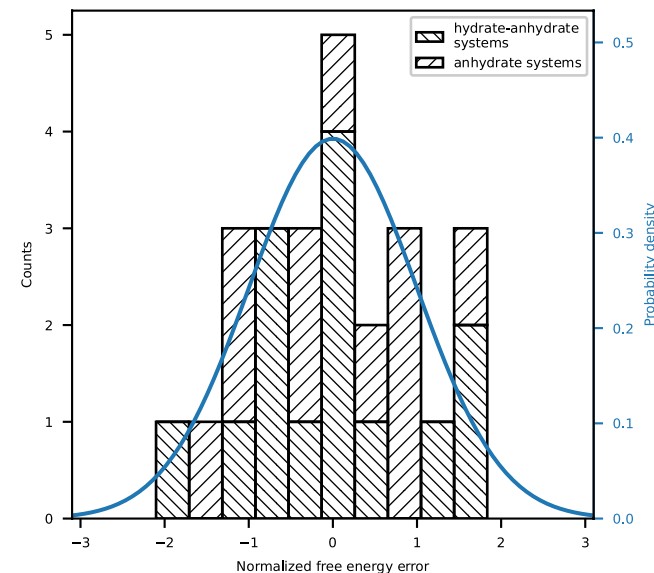

**Extended Data Fig. 2 | Distribution of normalised errors from both anhydrate and hydrate validation sets.** $(\Delta F - \Delta \hat{F})/\sigma_{\Delta F}$ is expected to have a mean of 0 and a standard deviation of 1 if $\Delta F - \Delta \hat{F}$ is normally distributed.

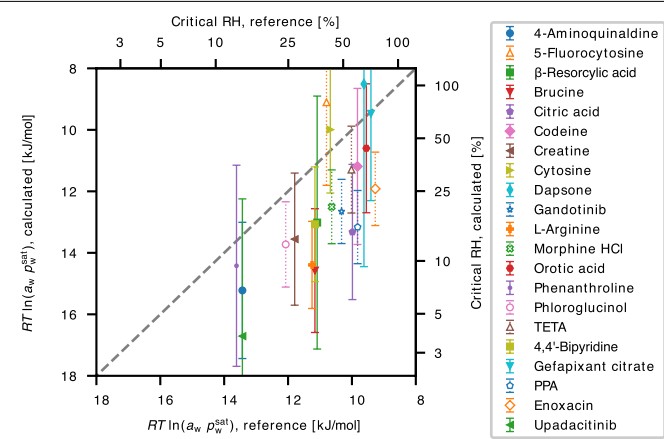

**Extended Data Fig. 3 | Calculated vs. reference pressure dependent part of the chemical potential of water at the phase transition without $\mu^{\circ}_{H_2O,corr}$.** The corresponding relative humidity is shown on the secondary axes. Reference systems excluded from the calculation of statistical errors due to ambiguities in experimental data are shown with open markers and dotted error bars. Error bars represent one standard error.

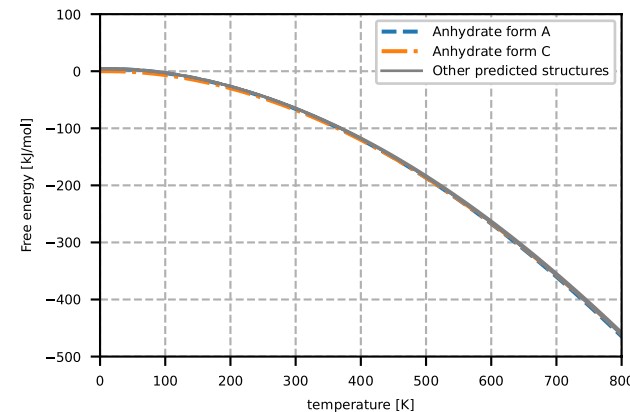

**Extended Data Fig. 4 | Absolute free energy vs. temperature for the anhydrate forms of radiprodil.** The free energy of the anhydrate form C at 0 K was set to zero.

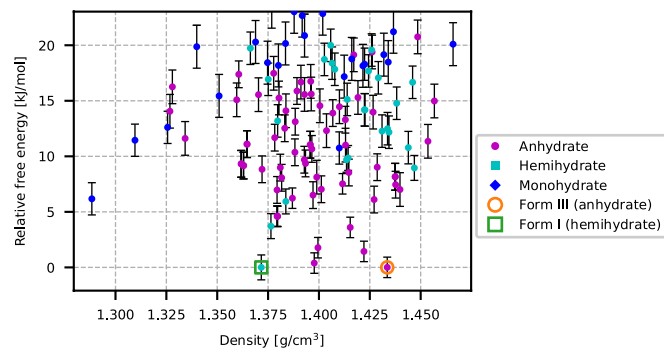

**Extended Data Fig. 5 | Free energy landscape of the predicted structures of upadacitinib.** Free energy landscape is shown at a temperature of 298.15 K and at a relative humidity of 7.8%, the predicted anhydrate-hemihydrate coexistence conditions.

**Extended Data Table 1 | Anhydrate systems used for validation**

| Compound | Validation type | Number of atoms | $\sigma_{\Delta F}$ [kJ/mol] | $\Delta F - \Delta \hat{F}$ [kJ/mol] |
|---|---|---|---|---|
| Acemetacin | Solubility | 47 | 1.85 | -0.902 |
| Acetazolamide | Solubility | 19 | 1.18 | -1.54 |
| Acetohexamide | Solubility | 42 | 1.75 | 1.38 |
| Famotidine | Solubility | 35 | 1.60 | -1.74 |
| Indometacin | Solubility | 41 | 1.41 | -0.00835 |
| Ritonavir[*] | Solubility | 98 | 2.68 | 0.805 |
| Rotigotine[*†] | Solubility | 47 | 1.85 | 1.07 |
| Verubecestat[*] | Solubility | 45 | 1.81 | -3.48 |
| Para-amino benzoic acid (PABA) | Solubility | 17 | 0.97 | 0.987 |
| Veliparib | Solubility | 34 | 1.58 | -2.42 |
| Radiprodil | Solubility | 49 | 1.89 | -0.983 |
| Paracetamol | Solubility | 20 | 1.21 | 1.207 |
| Gaboxadol HCl | Phase transition | 20 | 1.05 | 1.64 |
| Etiracetam | Phase transition | 26 | 1.38 | 0.496 |
| Diflunisal[*] | Phase transition | 26 | 1.19 | 0.0783 |
| Omarigliptin[‡] | Phase transition | 47 | 1.61 | 0.570 |

$\sigma_{\Delta F}$ represents one standard deviation of the error for the free energy difference between two crystal structures. $\Delta F - \Delta \hat{F}$ is the deviation between the calculated and the experimental free energy difference. Further information about experimental data and validation results can be found in the Supplementary Information.

[*]Experimental structure contains disorder.

[†]Experimental solubility values are not measured at infinite dilution conditions.

[‡]Experimental phase transition does not have measured upper and lower bounds.

**Extended Data Table 2 | Hydrate systems used for validation**

| Compound | Number of atoms in compound | Number of waters per compound in hydrate | $\sigma_{\Delta F}$ [kJ/mol] | $\Delta F - \Delta\hat{F}$ [kJ/mol] |
|---|---|---|---|---|
| 4-Aminoquinaldine | 22 | 1 | 1.42 | 0.0237 |
| 5-Fluorocytosine* | 13 | 1 | 1.24 | -3.46 |
| β-Resorcylic acid | 17 | 0.5 | 1.16 | 0.0791 |
| Brucine | 55 | 2 | 2.16 | 3.30 |
| Citric acid | 21 | 1 | 1.40 | 1.57 |
| Codeine | 43 | 1 | 1.89 | -0.405 |
| Creatine | 18 | 1 | 1.31 | -0.00716 |
| Cytosine | 13 | 1 | 1.17 | -2.45 |
| Dapsone | 29 | 0.33 | 1.01 | -0.960 |
| Gandotinib | 58 | 4 | 2.43 | 2.24 |
| L-Arginine | 26 | 2 | 1.50 | 2.75 |
| Morphine HCl[†] | 42 | 3 | 2.07 | 0.300 |
| Orotic acid | 15 | 1 | 1.23 | -0.718 |
| Phenanthroline | 22 | 1 | 1.43 | -0.962 |
| Phloroglucinol* | 15 | 2 | 1.39 | -0.0705 |
| Triethylenetetramine dihydrochloride (TETA)[†] | 32 | 2 | 2.26 | -0.982 |
| 4,4'-Bipyridine | 20 | 2 | 1.54 | 0.285 |
| Gefapixant citrate | 64 | 1 | 2.26 | -1.70 |
| Pipemidic acid (PPA)[†] | 39 | 3 | 2.02 | 3.706 |
| Enoxacin[†‡] | 40 | 3 | 2.04 | 2.642 |
| Upadacitinib | 46 | 0.5 | 1.45 | 0.755 |

$\sigma_{\Delta F}$ represents one standard deviation of the error for the free energy difference between two crystal structures. $\Delta F - \Delta\hat{F}$ is the deviation between the calculated and the experimental free energy difference. Further information about experimental data and validation results can be found in the Supplementary Information.
*Experimental structure contains disorder.
[†]Reference critical water activity is not reversibly determined.
[‡]Experimental structure was corrected based on calculations, further details are provided in the Supplementary Information.