## [Peer Review File · Nature]

Manuscript Title: Predicting crystal form stability under real-world conditions

Reviewer Comments & Author Rebuttals

Reviewer Reports on the Initial Version:

Referees' comments:

Referee #1 (Remarks to the Author):

This paper describes efforts to advance computational organic crystal structure prediction substantially closer to "real-world conditions" by predicting finite-temperature free energies and phase transition temperatures with statistically estimated uncertainties, and also predicting hydrate formation as a function of relative humidity. This isn't the first work to compute free energies & phase transitions, but the careful treatment of uncertainties and validation against experiment stands out sharply here. That information greatly increases the experimental utility of the predictions by providing the user with important context for the purported stabilities of any predicted forms, especially if they have not yet been discovered experimentally. The humidity-dependent predictions of hydrate formation are also very nice and address a major real-world issue in pharmaceutical solid formulations. I agree with the authors that these advances greatly increase the value of organic crystal structure prediction to experimentalists. The careful curation of so much experimental data for many pharmaceutical and other interesting molecules is a beautiful secondary benefit of this work.

Overall, the science underlying this work is very well done and has potentially broad implications for how crystal structure prediction is used by the pharmaceutical industry and others.

Unfortunately, I do not find the paper suitable for publication in Nature. The presentation is too technical and lacks a clear, unifying narrative that would appeal to the non-specialist audience of the journal. The transitions between sub-sections are very jarring and the sections almost lack continuity. I found the manuscript difficult to read even as an expert in the field, and I doubt the general audience reader would gain much insight from it at all. If anything, the paper reads more an internal research report than an outward facing publication.

I also am disappointed by the lack of physical insights communicated by the paper. I am left without a clear sense of how much the many different ingredients in their overall model (e.g. Fig 1) contribute. That makes it difficult to understand what features matter or to assess where future research efforts should focus. They do have a "Comparison of energy methods" section in the SI that seemingly tries to disentangle the various contributions, but again it reads more like a raw data report and lacks any clear narrative to help the reader disentangle the data and gain physical insights.

Referee #2 (Remarks to the Author):

The manuscript describes the development of methods for predicting crystal structures of organic molecules, focussing on pharmaceutical molecules and their hydrates. The work claims several advances, which are described as "1) further improving the accuracy of free energy calculations, 2) placing both anhydrate crystal structures and hydrate crystal structures of different stoichiometries on the same energy landscape, and 3) quantifying statistical errors for the computed free energies, which enables evaluation of the reliability of crystal structure prediction

and energy ranking results." These are all addressed in the manuscript and I see no flaws in the developments that are presented.

This is a nice study. In particular, it is valuable for the community to have a benchmark set of curated energy differences between crystal structures. These offer a set against which developments in the field can be evaluated. The comparison of anhydrate and hydrate crystal structures is also nice, although the authors do not place this within the context of what has already been done in this area – see below.

Overall, work in this area is naturally fairly incremental now that there are methods that can be used successfully for crystal structure prediction. In particular, the successful prediction by one of the authors of this manuscript of all crystal structures in the fourth blind test of crystal structure prediction (*Acta Cryst.* (2009). B65, 107-125) already demonstrated that these methods are valuable alongside experiments. The statements in this manuscript (line 43) "These contributions bridge the gap between experimental and computational data, transforming crystal structure prediction into a reliable and actionable procedure that can be used in combination with experimental insights to direct crystal form selection" (and a similar statement on line 68) do not really reflect the state of the field. It is already the case that crystal structure prediction is an actionable procedure. There are quite a few examples in the literature where new solid forms have been discovered after computational guidance based on CSP. Examples can be given in pharmaceutical materials science (for example, references 20, 21, 31), where the calculations gave guidance on the conditions for producing the new polymorph, and other areas, such as porous molecular crystals (eg. *Nature*, 543, 657–664 (2017)). Yes, the confidence in making decisions will improve as the accuracy of free energy differences is increased, but it is already the case that these methods are used alongside, and to guide, experiments.

Much of the method, summarized in Extended Data Figure 1, brings together existing methods (from references 25, 28, 37). I don't see much new in how the lattice energies of predicted crystal structures are calculated. The methods bring together some excellent developments that have been reported in these previous publications. Thus, the new developments reported here seem to relate to the phonon calculations: the treatment of soft modes, methyl group rotation, H-stretch anharmonicity and the "large supercell correction", which is included the sample the dispersion of acoustic phonons. However, I do not see where the authors show the impact of the additional calculations beyond the harmonic phonon calculations on accuracy of predicted free energy differences. The Supplementary Information shows the impact of removing the whole of the vibrational energy contribution to free energies (Table S5). This does not show whether the added calculations (ie, what is new in this paper) are a significant advance.

The methods described in this manuscript for comparing anhydrate and hydrate predicted crystal structures are a development beyond what has been demonstrated before, by including temperature and relative humidity in the comparison of stabilities of different stoichiometries. However, the statement (line 63): "Furthermore, CSP has thus far been limited to fixed stoichiometries" is not true. It is true that prediction of stoichiometry is not often included in crystal structure prediction studies. However, a few examples from the literature show that this aspect of CSP has been addressed, sometimes successfully:

"Towards Prediction of Stoichiometry in Crystalline Multicomponent Complexes". Cruz-Cabeza et al, *Chem. Eur. J.*, 14: 8830-8836

"Predicting stoichiometry and structure of solvates", Cruz-Cabeza et al, *Chem. Commun.*, 2010,46, 2224-2226

"Which, if any, hydrates will crystallise? Predicting hydrate formation of two dihydroxybenzoic acids", Braun et al, *Chem. Commun.*, 2011,47, 5443-5445 (which states "A study of two dihydroxybenzoic acid isomers shows that computational methods can be used to predict hydrate

formation, the compound:water ratio and hydrate crystal structures.")

CSP has also been used to predict cocrystallization, which is a related problem.

Beyond the technical developments that are presented, the authors define the term "physicochemical accuracy" as an error of 1 kJ/mol, which they distinguish from the often-quoted "chemical accuracy" of 1 kcal/mol and claim to show that free energy difference calculations between crystal forms are approaching this accuracy. In my opinion, this definition is less useful. Setting the value for this new definition as 1 kJ/mol feels arbitrary, apart from having a value of 1 in a smaller unit (kJ/mol) than the units often used for "chemical accuracy" (kcal/mol). It is certainly strange to define a target accuracy and claim that it is (almost) reached in the same publication. I would be much happier if practitioners in the field quoted the errors in the quantities that are of interest. If we are interested in predicting a transition temperature, then calculate the errors of that prediction and decide if those errors are sufficiently small for the application of the prediction. In the case of transition temperatures, an accuracy quite a bit better than 1 kJ/mol will often be required. The predicted range for transition temperature between forms A and C of one of the example compounds in this study (radiprodil) is about 250 K (Extended Data Figure 5 and Figure 3.) This is not good enough to guide experiments and does not meet the stated criterion (line 267) "If CSP is to be fully incorporated into industrial processes in, e.g., pharmaceutical development, properties such as temperature and relative humidity of phase transitions must be predicted with an accuracy similar to or better than experiment."

Overall, this is an excellent technical paper. It shows some of the best methods that currently exist for predicting crystal structures of organic molecules. I believe that it will be of interest to people using these calculations. However, the work does not present a big breakthrough in the field. Rather, it carefully examines the accuracy that is achieved by bringing together the best existing methods. The benchmark set of free energy differences will be useful for the community.

Referee #3 (Remarks to the Author):

Review of the manuscript Predicting crystal form stability under real-world temperature and humidity conditions by Firaha and co-workers.

The manuscript by Firaha et al describes a very well-thought and thorough piece of work that approaches a full mapping of the crystal free energy landscape of drug molecules, even considering the formation of hydrates. The drug molecules investigated are of size and complexity similar to present-day 'small molecule' drug targets, and thus of high industrial relevance.

Building on years of gradual improvements (in many cases by some of the authors of this manuscript) in methodology of crystal structure prediction, this manuscript presents the state of the art in a very convincing way. Furthermore, the work brings some interesting new approaches, in particular an approach to assess the uncertainty of the predicted free energies. This error estimate reveals that the present state of the art can assess free energy differences down to 1-2 kJ/mol, which is impressive. At the same time, calculated solid state transition temperatures and phase diagrams highlight that even these very small uncertainties translate into rather large uncertainties on such derived properties.

On top of demonstrating the excellent performance of this approach, the authors provide a benchmark set of experimental data.

The data and methods are well-described, or with clear references to the crystal structure prediction methodologies that have been published earlier.

The manuscript is overall well-written and easy to read.

Some questions that I miss being addressed by the authors are:

The 'per atom' estimate of free energy uncertainties is of course a very crude approach, as also noted by the authors. Do the authors consider these estimates to present a lower or a higher bound on the free uncertainties?

The lattice dynamical contribution to the free energies should in principle include an estimate of the acoustic phonons. From the description in the manuscript, it seems like only gamma-point frequencies were calculated. How were the acoustic phonons considered?

Author Rebuttals to Initial Comments: Reply to referees

We thank all three referees for their valuable comments that have resulted in substantial improvements of the manuscript.

The referees' comments are reproduced below and we have inserted our replies where appropriate.

We have also corrected some errors and made some additional improvements to the manuscript that were not requested by the referees. These changes are described at the bottom of this document.

Referee #1 (Remarks to the Author):

This paper describes efforts to advance computational organic crystal structure prediction substantially closer to "real-world conditions" by predicting finite-temperature free energies and phase transition temperatures with statistically estimated uncertainties, and also predicting hydrate formation as a function of relative humidity. This isn't the first work to compute free energies & phase transitions, but the careful treatment of uncertainties and validation against experiment stands out sharply here. That information greatly increases the experimental utility of the predictions by providing the user with important context for the purported stabilities of any predicted forms, especially if they have not yet been discovered experimentally. The humidity-dependent predictions of hydrate formation are also very nice and address a major real-world issue in pharmaceutical solid formulations. I agree with the authors that these advances greatly increase the value of organic crystal structure prediction to experimentalists. The careful curation of so much experimental data for many pharmaceutical and other interesting molecules is a beautiful secondary benefit of this work.

Overall, the science underlying this work is very well done and has potentially broad implications for how crystal structure prediction is used by the pharmaceutical industry and others.

Unfortunately, I do not find the paper suitable for publication in Nature. The presentation is too technical and lacks a clear, unifying narrative that would appeal to the non-specialist audience of the journal. The transitions between sub-sections are very jarring and the sections almost lack continuity. I found the manuscript difficult to read even as an expert in the field, and I doubt the general audience reader would gain much insight from it at all. If anything, the paper reads more an internal research report than an outward facing publication.

Reply 1.1

We have taken this comment very serious and entirely rewritten the manuscript.

In the abstract and the summary, we clearly mention the four main advances which are then described in more detail in the first four sections of the new manuscript: "Improved composite free energy calculations", "First-of-a-kind free energy benchmark", "Transferable error estimation" and "Hydrate-anhydrate phase transitions". In the fifth section, "Pharmaceutical case studies", we demonstrate the impact of our advances on modern crystal structure prediction. The subsequent "Discussion" and "Outlook" have also been completely rewritten to incorporate various remarks of all three referees. In general, we have shifted

technical detail from the main manuscript to the Supporting Information and spend more words on explaining the philosophy and the consequences of our approach.

I also am disappointed by the lack of physical insights communicated by the paper. I am left without a clear sense of how much the many different ingredients in their overall model (e.g. Fig 1) contribute. That makes it difficult to understand what features matter or to assess where future research efforts should focus. They do have a "Comparison of energy methods" section in the SI that seemingly tries to disentangle the various contributions, but again it reads more like a raw data report and lacks any clear narrative to help the reader disentangle the data and gain physical insights.

Reply 1.2

We are addressing this criticism in various ways:

- We explain the philosophy of our approach in section "Improved composite free energy calculations".
- We document how the overall results change when various energy components are disregarded in the Supplementary Information (see section "Comparison of energy methods" and in particular Table S4).
- We provide an Excel file, energy_components.xlsx, with detailed energy components for the crystal structures of the benchmark as Supplementary Material.
- In the "Outlook" we discuss potential directions for further improvement.

We explain in the Supplementary Information (see first paragraph of "Comparison of energy methods") that the number of samples is insufficient to conclude if the imaginary mode correction, the very soft mode correction, the methyl top correction and the hydrogen bond anharmonicity correction are beneficial, or maybe even detrimental. The impossibility to conclude is now indicated by additional question marks in Extended Data Figure 1.

Referee #2 (Remarks to the Author):

The manuscript describes the development of methods for predicting crystal structures of organic molecules, focussing on pharmaceutical molecules and their hydrates. The work claims several advances, which are described as "1) further improving the accuracy of free energy calculations, 2) placing both anhydrate crystal structures and hydrate crystal structures of different stoichiometries on the same energy landscape, and 3) quantifying statistical errors for the computed free energies, which enables evaluation of the reliability of crystal structure prediction and energy ranking results." These are all addressed in the manuscript and I see no flaws in the developments that are presented.

This is a nice study. In particular, it is valuable for the community to have a benchmark set of curated energy differences between crystal structures. These offer a set against which developments in the field can be evaluated. The comparison of anhydrate and hydrate crystal structures is also nice, although the authors do not place this within the context of what has already been done in this area - see below.

Overall, work in this area is naturally fairly incremental now that there are methods that can be used successfully for crystal structure prediction. In particular, the successful prediction by one of the authors of this manuscript of all crystal structures in the fourth blind test of crystal

structure prediction (Acta Cryst. (2009). B65, 107-125) already demonstrated that these methods are valuable alongside experiments. The statements in this manuscript (line 43) "These contributions bridge the gap between experimental and computational data, transforming crystal structure prediction into a reliable and actionable procedure that can be used in combination with experimental insights to direct crystal form selection" (and a similar statement on line 68) do not really reflect the state of the field. It is already the case that crystal structure prediction is an actionable procedure. There are quite a few examples in the literature where new solid forms have been discovered after computational guidance based on CSP. Examples can be given in pharmaceutical materials science (for example, references 20, 21, 31), where the calculations gave guidance on the conditions for producing the new polymorph, and other areas, such as porous molecular crystals (eg. Nature, 543, 657-664 (2017)). Yes, the confidence in making decisions will improve as the accuracy of free energy differences is increased, but it is already the case that these methods are used alongside, and to guide, experiments.

Reply 2.1:

We agree and have replaced „a reliable and actionable procedure“ by “a more reliable and actionable procedure“. However, we would like to stress that we have used the methods described in our manuscript in commercial CSP studies for almost 2 years now and that they truly make a huge difference to the interpretability of CSP studies.

Much of the method, summarized in Extended Data Figure 1, brings together existing methods (from references 25, 28, 37). I don't see much new in how the lattice energies of predicted crystal structures are calculated. The methods bring together some excellent developments that have been reported in these previous publications. Thus, the new developments reported here seem to relate to the phonon calculations: the treatment of soft modes, methyl group rotation, H-stretch anharmonicity and the "large supercell correction", which is included the sample the dispersion of acoustic phonons. However, I do not see where the authors show the impact of the additional calculations beyond the harmonic phonon calculations on accuracy of predicted free energy differences. The Supplementary Information shows the impact of removing the whole of the vibrational energy contribution to free energies (Table S5). This does not show whether the added calculations (ie, what is new in this paper) are a significant advance.

Reply 2.2

We have already addressed this point in reply 1.2 to referee #1.

The methods described in this manuscript for comparing anhydrate and hydrate predicted crystal structures are a development beyond what has been demonstrated before, by including temperature and relative humidity in the comparison of stabilities of different stoichiometries. However, the statement (line 63): "Furthermore, CSP has thus far been limited to fixed stoichiometries" is not true. It is true that prediction of stoichiometry is not often included in crystal structure prediction studies. However, a few examples from the literature show that this aspect of CSP has been addressed, sometimes successfully:

"Towards Prediction of Stoichiometry in Crystalline Multicomponent Complexes". Cruz-Cabeza et al, Chem. Eur. J., 14: 8830-8836

"Predicting stoichiometry and structure of solvates", Cruz-Cabeza et al, Chem. Commun., 2010,46, 2224-2226

"Which, if any, hydrates will crystallise? Predicting hydrate formation of two dihydroxybenzoic acids", Braun et al, Chem. Commun., 2011,47, 5443-5445 (which states "A study of two dihydroxybenzoic acid isomers shows that computational methods can be used to predict hydrate formation, the compound : water ratio and hydrate crystal structures.")

CSP has also been used to predict cocrystallization, which is a related problem.

Reply 2.3

We present our apologies for having made a claim that was too strong. The three suggested references are cited in the manuscript as follows: "Furthermore, CSP has been applied to the prediction of the stoichiometric hydrates³⁷ and solvates,^{38,39} but without explicitly considering relative humidity or solvent activity."

Beyond the technical developments that are presented, the authors define the term "physicochemical accuracy" as an error of 1 kJ/mol, which they distinguish from the often-quoted "chemical accuracy" of 1 kcal/mol and claim to show that free energy difference calculations between crystal forms are approaching this accuracy. In my opinion, this definition is less useful. Setting the value for this new definition as 1 kJ/mol feels arbitrary, apart from having a value of 1 in a smaller unit (kJ/mol) than the units often used for "chemical accuracy" (kcal/mol). It is certainly strange to define a target accuracy and claim that it is (almost) reached in the same publication. I would be much happier if practitioners in the field quoted the errors in the quantities that are of interest. If we are interested in predicting a transition temperature, then calculate the errors of that prediction and decide if those errors are sufficiently small for the application of the prediction. In the case of transition temperatures, an accuracy quite a bit better than 1 kJ/mol will often be required. The predicted range for transition temperature between forms A and C of one of the example compounds in this study (radiprodil) is about 250 K (Extended Data Figure 5 and Figure 3.) This is not good enough to guide experiments and does not meet the stated criterion (line 267) "If CSP is to be fully incorporated into industrial processes in, e.g., pharmaceutical development, properties such as temperature and relative humidity of phase transitions must be predicted with an accuracy similar to or better than experiment."

Reply 2.4

We agree and have dropped the concept of physico-chemical accuracy altogether. We saw physico-chemical accuracy as some point of entry level to predicting phase transitions but agree that it is more appropriate to discuss which energy accuracy is required to reach a certain target accuracy for phase transition temperatures and other measurable quantities. In the discussion we now write: "At the current level of accuracy, hydrate-anhydrate phase transition relative humidities are predictable to within a factor of 1.7 and the 1σ error of 183 K obtained for the anhydrate-anhydrate phase transition temperature of radiprodil is representative for what has been observed in numerous confidential contract research studies. Hence, at present our method enables the prediction of likely phase transitions between two crystal forms, though not at which point it will occur. The prediction of phase transition temperatures to within 10 K will require a further improvement of the accuracy by a challenging factor of 20."

The change of perspective has also substantially affected the outlook. In the very last sentence, we pose the calculation of more accurate lattice free energies as a challenge to the scientific community.

Overall, this is an excellent technical paper. It shows some of the best methods that currently exist for predicting crystal structures of organic molecules. I believe that it will be of interest to people using these calculations. However, the work does not present a big breakthrough in the field. Rather, it carefully examines the accuracy that is achieved by bringing together the best existing methods. The benchmark set of free energy differences will be useful for the community.

Reply 2.5

There are some major accuracy improvements that are hopefully better documented now that we have added additional information to the section „Comparison of energy methods” of the SI (see Table S 4). To give one example: Adding the single molecule MP2D correction to what is generally called PBE0+MBD+ F_{vib} increases the accuracy for anhydrate-anhydrate free energy differences by 30%. Please note that you never know if a correction supposed to be additive effectively works until you have really tried.

Also, the force field large-cell correction works very well and helps to save substantial amounts of CPU time by limiting ab initio phonon calculations to very small super cells. Overall, these improvements make a big difference to performance.

Referee #3 (Remarks to the Author):

Review of the manuscript Predicting crystal form stability under real-world temperature and humidity conditions by Firaha and co-workers.

The manuscript by Firaha et al describes a very well-thought and thorough piece of work that approaches a full mapping of the crystal free energy landscape of drug molecules, even considering the formation of hydrates. The drug molecules investigated are of size and complexity similar to present-day 'small molecule' drug targets, and thus of high industrial relevance.

Building on years of gradual improvements (in many cases by some of the authors of this manuscript) in methodology of crystal structure prediction, this manuscript presents the state of the art in a very convincing way. Furthermore, the work brings some interesting new approaches, in particular an approach to assess the uncertainty of the predicted free energies. This error estimate reveals that the present state of the art can assess free energy differences down to 1-2 kJ/mol, which is impressive. At the same time, calculated solid state transition temperatures and phase diagrams highlight that even these very small uncertainties translate into rather large uncertainties on such derived properties. On top of demonstrating the excellent performance of this approach, the authors provide a bench-mark set of experimental data.

The data and methods are well-described, or with clear references to the crystal structure prediction methodologies that have been published earlier.

The manuscript is overall well-written and easy to read. Some questions that I miss being addressed by the authors are:

The 'per atom' estimate of free energy uncertainties is of course a very crude approach, as also noted by the authors. Do the authors consider these estimates to present a lower or a higher bound on the free uncertainties?

Reply 3.1

The referee raises an important question. In the discussion we now write: “Comparing the magnitudes of the errors σ_{at} and σ_{H_2O} , we see that the error per atom of the water molecule at 0.379 kJ/mol is approximately two times larger than the error per non-water atom at 0.191 kJ/mol. Both values are affected by the computational and the experimental error, and as such represent upper limits for the actual computational error. The larger error per atom for water is consistent with the fact that water is generally considered difficult to model. On the contrary, 70% of the atoms of the compounds in our test set are carbon atoms or their hydrogen bonded neighbours that are much easier to describe. The factor of two difference between the two errors suggests a substantial dependence of the error per atom on the atomic species. Therefore, the derived value for σ_{at} is an average that strictly speaking only applies to compounds that are close to the average chemical composition of the benchmark.”

Please see also the related correction mentioned further down.

The lattice dynamical contribution to the free energies should in principle include an estimate of the acoustic phonons. From the description in the manuscript, it seems like only gamma-point frequencies were calculated. How were the acoustic phonons considered?

Reply 3.2

To clarify the issue of the calculations of acoustic modes the description of the large-cell correction in the Method section has been rewritten: “To conserve CPU time, phonon calculations at ab initio level are limited to very small supercells that are not large enough to capture the effect of phonon band structure on the lattice free energy, in particular for acoustic modes. Therefore, a large cell correction was carried out using tailor-made force fields⁷⁰ reparametrized with additional reference data at the PBE-NP level of theory for the forces calculations. Using the phonon calculations described above including imaginary mode and soft mode corrections, force field lattice free energies were computed for the small supercell already used in the ab initio phonon calculations and a larger supercell with minimal distances of 24 Å between symmetry copies of an atom. The difference of the two lattice free energy calculations was used as a correction. As already described above, eigenvalues and eigenmodes were explicitly calculated for every k-point compatible with the periodic boundary conditions of the supercell. This way the band structure of all modes, including acoustic modes, is explicitly taken into account. The contribution of the three acoustic modes at the gamma-point was neglected. It is important to note that imaginary modes can only be explicitly sampled and evaluated if an explicit supercell is available for the corresponding k-point.”

Additional changes not requested by the referees

Correction 1: When preparing the energy information table for Reply 1.1, we noticed that a statement in the original manuscript was wrong. In two of the small cell ab initio phonon calculations imaginary modes are actually observed. We previously wrote that there were no imaginary modes. The manuscript has been corrected accordingly.

Correction 2: Modifying the manuscript, we noticed that Form A and one predicted structure of radiprodil were flipped to compute single point and single molecule corrections energies. These energies were combined with correct free energy for some of the figures. For the same reason, a wrong space group and Z' were mentioned for Form A of radiprodil anhydrate in the first version of the manuscript, but were correct in the Supporting Information. In the revised manuscript and the SI, we fixed these inconsistencies.

Correction 3: When drafting Reply 3.1, we noticed that the error per atom in water was previously wrongly calculated from the error per water. One should divide by $\sqrt{3}$, not by three as previously done, this mistake has now been corrected.

Correction 4: The previously show error for the phase transition temperature in Figure 4 was too small. Figure 4 has been corrected such that the error bar is consistent with Eq. S 21 of the Supplementary Information.

Improvement 1: The discussion of previous hydrate anhydrate work in the Summary has been complemented by two further references: "Free energy calculations have been used to construct hydrate-anhydrate phase diagrams, still requiring experimental calibration for every compound and pair of crystal forms.⁴⁰ A data-driven and topological algorithm⁴¹ has been recently employed in conjunction with CSP to the prediction of fractional or nonstoichiometric hydrates to evaluate the ever-present risk of hydrate formation for industrially relevant compounds".

Improvement 2: Since the first submission, a name has been agreed on for our improved composite energy calculation method. This name is now introduced in the Method section as follows: "The name of our energy calculation method, TRHu(ST) 23 is an acronym that stands for "T_{em}perature and R_{elative} H_{umidity} dependent free energy calculations with S_Tandard deviations" and should be pronounced "Trust 23". The name is meant to encompass both the actual energy calculations and the model for transferable error estimates calibrated on a specific benchmark. Because the energy calculations, the error estimation model and the benchmark will continue to evolve, we add the year of publication to refer to our very first implementation and to define a naming scheme for improvements to come."

Improvement 3: Figures have been shifted between the main manuscript, the extended data and the Supporting Information to improve the readability of the manuscript.

Improvement 4: We added to the Supplementary Material PBE-NP minimized structures for radiprodil and upadaticitinib landscapes presented in the manuscript and the Supplementary Information. Also, experimental and PBE-NP minimized structures of omarigliptin Form 1, radiprodil monohydrate and dihydrate were added to the Supplementary Material.

Reviewer Reports on the First Revision:

Referees' comments:

Referee #1 (Remarks to the Author):

The revised paper reads much better than the original submission, telling a clearer story. The authors have also improved the discussion of the importance of different contributions to the model more thoroughly.

Nevertheless, along the lines of my initial comments (and those of Referee #2), I still feel this article is probably too technical and specialised for the broad audience of Nature. I base this comment on two criteria:

(1) Is there a major scientific breakthrough?

I continue to be impressed by the overall quality of their modeling. As Referee #2 correctly notes, the technical advances here largely represent combining a long list of ideas that have been developed by the community over the years, including contributions from these authors themselves. The work really highlights how far one can go by paying careful attention to as many different details as possible. The results here are almost certainly of real benefit to pharmaceutical companies and others working with organic crystals. The benchmark data set they have curated here is also very nice. At the same time, I don't see any single breakthrough that is transformationally different from what others are doing in the space, at least at the level of a Nature publication. The methodological differences here are more incremental.

(2) How much would a scientist entirely outside chemistry gain from reading the paper?

Such a reader would learn something about the field (and the revisions have definitely helped this), but not enough in my view. Much of the content is at a technical level that does not really speak to an audience outside its domain well.

Overall, the work is very good (but not outstanding) for criterion #1, and fair on criterion #2. From that perspective, publishing this work in one of the more chemistry-focused Nature-family journals such as Nature Chem, etc, would probably be more appropriate.

Minor:

- With regards to the discussions of uncertainties and the errors in predicting phase transition temperatures, the following articles by Abramov et al are relevant and should probably be noted:

"Solid-Form Transition Temperature Prediction from a Virtual Polymorph Screening: A Reality Check" (DOI: 10.1021/acs.cgd.9b00989)

"Uncertainty Distribution of Crystal Structure Prediction" (DOI: 10.1021/acs.cgd.1c00527)

Referee #2 (Remarks to the Author):

My view on the revised manuscript is largely unchanged from my initial review.

The novel aspects of the work are: 1) the collection of free energy differences between solid forms and quantification of errors in free energy calculations and 2) the method for assessing anhydrous

and hydrate crystal structures as a function of conditions (temperature, relative humidity). Both of these contributions are valuable for the community.

The work also claims an improvement in free energy calculations. The results are excellent, but improvements here are small and largely a result of combining previously published methodologies. The points raised by the authors in reply to my original review are:

"Reply 2.5

There are some major accuracy improvements that are hopefully better documented now that we have added additional information to the section „Comparison of energy methods“ of the SI (see Table S 4). To give one example: Adding the single molecule MP2D correction to what is generally called PBE0+MBD+Fvib increases the accuracy for anhydrate-anhydrate free energy differences by 30%. Please note that you never know if a correction supposed to be additive effectively works until you have really tried.

Also, the force field large-cell correction works very well and helps to save substantial amounts of CPU time by limiting ab initio phonon calculations to very small super cells. Overall, these improvements make a big difference to performance."

The MP2D correction was proposed and demonstrated to improve CSP calculations in reference 32 and 33. The current work extends that testing by demonstrating that it systematically improves results on the set of free energies that have been compiled.

The large-cell phonon correction is shown to improve results and is a practical approach to combine force field and DFT phonons. It is a lower-cost approach than than full calculations at the DFT level (as in reference 25). Other low-cost approaches to the acoustic phonon contribution have been applied elsewhere (eg using a Debye model, ref 21).

I do not see any problems with the statistical treatment of errors and the reliability of the results.

I see this study as better suited to a less general journal, where the technical details and analysis of which components of the energy calculations make a statistical difference are in the main text. The table and figures in the SI that break down the various contributions are the most useful aspect of the combined method that is employed for free energy calculations.

Referee #3 (Remarks to the Author):

It is clear that the authors have taken the review comments from all reviewers into account in their rewritten manuscript. My comments have been clarified, and the manuscript has been significantly improved in its readability and in the discussion of results and future perspectives. I can recommend publication of the current version.

Author Rebuttals to First Revision:

Referee #1 requested to add a reference to "Solid-Form Transition Temperature Prediction from a Virtual Polymorph Screening: A Reality Check" (DOI: 10.1021/acs.cgd.9b00989). This article is now referenced in the Supporting Information where we derive our equation for the uncertainty of the phase transition temperature (see "Transition temperature error calculation" section):

"An alternative formula has been suggested and discussed for the example of two paracetamol polymorphs.¹⁰³"

Referee #1 also requested to add another reference. We do not believe that the other article, "Uncertainty Distribution of Crystal Structure Prediction" (DOI: 10.1021/acs.cgd.1c00527) is relevant in the context of our work. The article mainly describes comparisons of gas phase ab initio calculations to a gas phase ab initio gold standard method (CCSD(T)), also adding some extra solid state single point energy calculations which again compare low precision ab initio calculations to a high precision reference, which on top is different for the two chemical compounds that are considered. The work is very far away from our comparison of predicted free energies of fully energy optimized crystal structures to an experimental (not computed) reference that spans a large range of chemical compounds with a consistent approach. Since we have already reached the limit of 50 references for the main text, adding this reference would force us to remove another, more valuable reference. Also, we are unsure why and how the article should actually be cited. In case rejecting the additional reference is not an option, we would like to invite referee #1 to make a suggestion on where and how (not more than one extra short sentence to introduce the additional reference) the other article should be cited.